# “A ‘Cool’ Kid Wears a Brand, and Everyone’s following Him” Hierarchal Social Status in Preadolescence: A New Developmental Perspective

**DOI:** 10.3390/children11050547

**Published:** 2024-05-03

**Authors:** Hannah Fisher Grafy

**Affiliations:** Department of Psychology, Bar-Ilan University, Ramat Gan 5290002, Israel; hamania10@gmail.com

**Keywords:** hierarchal social status, preadolescence, socio-emotional development, latency stage, social status

## Abstract

Inequality in hierarchical social status, especially among socially excluded children, profoundly affects preadolescents. Historically viewed through a lens of psychopathology and moral deficiencies, it challenges the education system’s approaches and interventions. This article introduces a developmental perspective, highlighting the hierarchical social status‘ role in shaping classroom cohesion, strength, and distinctiveness. This study’s phenomenological, qualitative methodology aimed to gain preliminary insight into the children’s perspectives. Drawing from 12 focus group discussions involving 140 latency-age (grade 5) children in Israel, it uniquely reveals the dynamic nature of hierarchical social status influenced by children’s connections with the group. This dynamism promotes group unity, strengthens bonds, and prioritizes collective concerns, contributing to the development of a “social self” in the latency phase. Beyond theory, this study proposes innovative interventions to address social status disparities.

## 1. Introduction

Human societies inherently exhibit a propensity to arrange themselves into structured social hierarchies. Within these hierarchies, individuals aspire to ascend the ranks of their respective reference groups, recognizing that falling short in this endeavor can significantly affect their economic and psychological well-being [1,2,3]. A central theme when exploring social hierarchies revolves around how individuals identify themselves within social classes. In sociopsychological research, Huang [4] identified three prominent theoretical approaches to this subject. The first, the “realistic argument” originally formulated by foundational sociologists like Marx and Engels [5], posits that societal class divisions—particularly in industrialized societies—directly dictate the individuals’ subjective class identification. For instance, Marx’s perspective clearly delineated between the bourgeoisie and working classes, where objective conditions like owning the production means and relying on selling labor to the bourgeoisie inform individuals’ identification with the lower or upper class.

The second approach, the “reference group” perspective [6,7], emphasizes social comparison in shaping subjective class identification. In this view, the influence of significant others within a personal sphere mediates the connection between objective circumstances and an individual’s perceived class position. The third approach, the “reality and reference blend” [8], synthesizes elements of the previous two perspectives. It posits that class identification results from combining objective reality and subjective comparisons with reference groups. This perspective recognizes the interplay of sociopsychological forces driving centralization and socioeconomic forces leading to dispersion in class identification.

Like adults, children in the latency age group (9–12 years) also place importance on their social status as perceived by the children in their class. Several of the terms for social status in these peer relationships are rooted in psychology; others stem from the field of sociology: social preference (likeability [9] (p. 13)), social impact—perceived popularity (popularity [10]), sociometric status [11], sociometric popularity [12,13], social status (perceived status rather than actual interpersonal likes or dislikes, referring to relationships defined by honor, respect, admiration, deference, and reputation) [11,14,15]. Social status gives the children prestige, prominence, and respect within their social groups. In the school environment context, the hierarchal social status assumes particular significance for children. Factors such as acceptance, rejection, and number of friends are key indicators of the quality of the students’ social interactions during learning and recreational activities [16].

Hierarchal social status significantly affects children in different areas [17,18,19]. Emotionally, in the classroom, children view those with a high hierarchical social status as “cool” and appealing based on their attractiveness. This affects the children’s self-confidence, self-worth, and emotional, cognitive, and educational states [20,21,22]. In stark contrast, children from lower social classes often reside on the fringes or experience social exclusion. Their precarious social standing carries profound consequences that affect multiple dimensions of their current and future well-being, encompassing their physical [23,24,25,26,27] and emotional health [28,29,30], cognition [24,31], and adaptation [32].

The numerous powerful effects of social status on multiple developmental aspects seem to lead children to direct their focus onto hierarchal social status, and prioritize popularity above other considerations [19,21,33,34]. The literature has paid limited attention to understanding how a social status hierarchy influences children’s development during the latency stage. This study delves into the underlying dynamics that drive children of this age to direct their focus onto hierarchal social status.

Research examining social-status-related factors found that children differ in social status according to physical characteristics, such as strength, height, and athletic competence [35]. Studies also identified varying behavioral profiles among individuals of different social statuses (e.g., [36,37,38]). Newcomb et al.’s [39] meta-analysis, as well as Gifford-Smith and Brownell’s [40] and Lease et al.’s [22] reviews, provide detailed descriptions of these differences among sociometric groups. Based on their analyses, the key characteristics associated with each sociometric category are:Popular children are known for their superior prosocial abilities. Compared to the average children, they exhibit better social problem-solving skills, engage more frequently in enjoyable peer interactions, and display lower aggression levels;In contrast, rejected children tend to be more aggressive and disruptive and less sociable than average children;Although less aggressive, neglected children also are less sociable than average children. They participate in fewer social interactions and tend to be more withdrawn;Controversial children have a unique behavioral reputation. They are typically as sociable as popular children, but can be equally or more aggressive than rejected children. What sets them apart from rejected children is their ability to offset the negative effects of their aggressive behavior with advanced social skills.

These studies indicate a connection between social status and social skills. Highly popular children possess prosocial social skills; socially rejected children behave aggressively and lack prosocial skills. However, several studies have indicated opposing findings (e.g., [41,42,43,44]). Ethologists who study children’s peer groups emphasize the importance of social dominance in determining the group’s organizational structure [22]. Studies have shown that aggressiveness also characterizes children higher in the social hierarchy. These suggest that dominance hierarchies form in many social groupings, such that children at the top have more power and access to scarce and valuable resources than those at the bottom (e.g., [43,45,46]).

Aggressiveness gives rise to higher social status [47,48,49]. The children fight among themselves and use various strategies to obtain social dominance, power [43], and control over a group. Some strategies are coercive; others explicitly involve aggressive and threatening behavior [47,48,50,51,52,53,54]. Some use prosocial behavior and cooperative relationships with partners [52,55,56,57,58].

These theories frame hierarchal social status as a social structure stemming from motives, even basing it on struggles to gain power and control. However, these findings are contentious. According to Cillessen and Rose [13] (p. 102), the peer group’s social dynamics are neither clear nor understandable. In sum, the approaches in the literature regarding characteristics of different social statuses in the classroom point to three distinct needs: being popular, being liked, and having control. However, these approaches do not consider observing the classroom hierarchy through a psychological–developmental prism. Thus, this study aims to explore and gain a comprehensive psychological–developmental understanding of hierarchal social status during the latency phase.

Accordingly, the research questions are:(1)How do children of preadolescence (latency) age perceive the various social statuses (e.g., accepted, “nerdy”, “cool”, socially rejected) within their classroom hierarchy?(2)What can be derived from the children’s perspectives on the social statuses within their classroom’s hierarchy?

The researcher expected the study results to show that:(a)Children at the top of the social hierarchy are not motivated by personal desires or self-gain, but for the group’s welfare, cohesion, and unity;(b)The social hierarchy in preadolescence is dynamic and subject to change.

## 2. Materials and Methods

### 2.1. Approach

The researcher chose a phenomenological, qualitative approach to understand how children of latency age think and react toward hierarchal social status. At the phenomenological paradigm’s core is the notion that there is no objective reality. Rather, reality is interpreted through the eyes of those experiencing it. This perspective asserts that the participants’ subjective experiences enable a deep understanding of their unique perceptions and the cognitive “structures” guiding them [59,60]. It allows a broader interpretation to understand a particular phenomenon from several individuals’ perspectives.

Phenomenological research often presents intersubjective understanding as a unifying theme [61] (p. 2012). Due to the subject’s sensitivity for different children, the researcher chose a method that allowed for exploring and clarifying their perceptions and emotions about hierarchal social status while considering the beneficence principle—our ethical responsibility to act in others’ best interest. This responsibility encompasses safeguarding and advocating for the rights of others, preventing harm, mitigating conditions that might lead to harm, assisting individuals, and intervening to rescue individuals in unsafe situations [61]. Our choice of this paradigm stemmed from the lack of qualitative research documenting the group members’ subjective opinions, attitudes, and feelings about the subject [59,60].

### 2.2. Participants

The participants were 140 Jewish Israeli 5th-grade children (65 boys and 75 girls) aged 10 to 11 years from five elementary schools (economic status: middle class). In total, 12 classes participated, each considered a separate focus group (Table 1). Groups were gender-heterogeneous but differed in their demographic characteristics. Three groups were from religious schools, and nine were from secular schools. These groups align with the broader national demographic profile. The groups were small relative to the class size because children who did not bring consent forms signed by their parents could not participate in the study.

Focus groups are a valuable phenomenological research tool, providing several advantages over in-depth interviews. Among these advantages is the ability to gather information from a relatively large group of participants in a relatively short time. In this study, in-depth interviews were deemed inappropriate due to the paramount concern for safeguarding the children’s emotional well-being. Consequently, the researcher presented to the children a hypothetical scenario of social exclusion deliberately disconnected from their experiences.

The researcher deliberately selected a social exclusion scenario to trigger emotions tied to hierarchal social status, assuming that socially rejected children are at the lowest status. It was intended to evoke the children’s emotions and thoughts, enhancing the potential for a rich, engaging group discussion. Throughout all focus group sessions, the researcher ensured that the children refrained from sharing real-life incidents from their school experiences. However, some children inevitably recalled incidents from their classrooms. In such instances, the researcher emphasized that they should avoid disclosing specific individuals or names. Instead, the ongoing dialogue was guided toward general attitudes or perceptions without delving into specific classroom occurrences. The core of each focus group session revolved around discussing “other” children within a specially crafted story about children in another city:

In a fifth-grade class in another city, the students independently made decisions on many matters. One day, they decided to throw a party and sent invitations to everyone except for two children. The following day, the two children and their parents contacted the homeroom teacher to express their grievances about not receiving invitations. The teacher acknowledged the two students’ concerns and promptly contacted the parent committee, instructing the parents and students that all classmates must be invited. (It is common in Israel for school staff to address and act upon social exclusion, even for events outside of school).

The researcher read the story aloud. Initially, the children responded spontaneously and openly to the narrative. Following this, the researcher asked questions based on a protocol tailored specifically for this study: Why were neither of the two children invited? Who in the class made this decision? What might be the two uninvited children’s social status? What are the differences between the uninvited children and the other children in the class?

Drawing upon their responses, the researcher asked additional questions to delve deeper into the children’s perceptions of social status hierarchies within the group, as inferred from the narrative of the two children’s exclusion. The participants’ responses were audio-recorded during the focus group, written as general notes, and transcribed from the audio recordings [62]. The transcripts comprised 98 pages documenting 13 h of audio recordings.

### 2.3. Procedure

The focus groups took place during the school day and lasted between 60 and 90 min each. According to the Ministry of Education requirements, the homeroom teacher escorted the focus group participants and the researcher. The teacher sat at the back of the classroom and did not take part or intervene in the group discussion. Although some children were likelier than others to participate in a group setting, the researcher made a point to listen to the opinions of all the children (including the popular and the socially rejected children) and refrained from showing bias supporting the socially rejected children’s statements. Before or after each focus group session, the homeroom teacher provided the researcher with brief information about the speaking children’s social status.

### 2.4. Data Analysis

The data were analyzed using the Van Kaam method perfected by Moustakas [63]. The Moustakas process involves conducting a phenomenological study step-by-step, providing more in-depth insights into the results of other successful phenomenological studies from distinct contexts in psychology ([64] (p. 69), [65]). To enhance the findings’ reliability, a professional colleague of the researcher volunteered to be the second reader, and both reviewed all the dialogues in the focus groups. The process included seven stages. In the first stage, the two readers separately identified all statements specifying hierarchal social status. Second, all statements unrelated to hierarchal social status were excluded. Third, all relevant statements were classified into themes. In the fourth stage, each reader created a separate description of the themes for each focus group. In the following stages, the researcher united all themes from the two readers, identified three general central themes, and finally created a structural description that expresses the children’s attitudes regarding hierarchal social status and sheds light on how they perceive it.

### 2.5. Ethics

The Ethics Committees at [blinded for peer review] University and the Ministry of Education, Chief Scientist Department, approved this study. After receiving the approvals, potential schools were identified, and informed consent letters allowing their children to participate were sent for parents to sign. The letters specified the study’s purposes, guaranteed that the parents could refuse their children’s participation or leave the study at any stage, and assured anonymity. The focus group dates were set after the schools had gathered the parents’ consent letters.

Before each focus group, the homeroom teacher met with and informed the researcher about the class’s hierarchal social status. This information enabled the researcher to understand the context of each participant’s and group’s discourse and facilitate the group so that no children would be harmed or feel uncomfortable. Further, the names of the schools and the children who participated were omitted when transcribing the materials. Omitting the names ensured the participants’ privacy and allowed the researcher to remain neutral when analyzing the data.

## 3. Results

The thematic analysis of participants’ responses to the research story yielded three overarching themes: (a) Accepted children are better friends, (b) a “cool” kid wears a brand, and everyone’s following him, and (c) I made the biggest change in my life.

### 3.1. “Accepted Children Are Better Friends”

The initial theme that emerged from the children’s statements about differences in social status highlights the children’s tendency to prioritize the group. Socially accepted peers were described as individuals who placed significant importance on the group, actively participated in it, and demonstrated heightened awareness of the group’s central role in their thoughts and actions. They showed a propensity for social interactions, whereas those who were less accepted seemed to maintain a more distant connection to the group.

“There are those, the ‘accepted’ ones, who seem to be more into social matters and there are those who are shyer to talk and don’t join in playing with us and just sit on the sidelines.” In contrast to the socially accepted children, who were deeply embedded within the social fabric and vested in societal interactions, the less accepted children appeared more self-focused and preoccupied with personal concerns.

“Maybe when the popular kids are playing outside during the break, the not-so-popular kids tend to seclude themselves and stay alone in the classroom, only mind their own business.” Children in the focus groups distinguished between socially accepted and less connected peers in their engagement during activities. Whereas the socially accepted children were outside the classroom playing group games during breaks, the less popular children remained inside, occupied with their own interests. Additionally, another distinction emerged. The children perceived the socially accepted children as better friends who readily assist their peers and the less popular children as reluctant to offer needed help.

“They (the accepted children) are better friends…. Let’s say he asks you for help, so you say, ‘I will gladly help you.’ And they (the unaccepted children) say, ‘no’.” According to the children’s perceptions, friendly behavior requires prosocial social skills. In addition, the children must obey the group, comply with group norms, and accept societal authority like a soldier obeying a commander with discipline and obedience.

Kids in the class tell other kids to bring sweets or other things, but they (the unaccepted children) don’t listen to them and don’t bring what they were asked to bring…. After that, they won’t be invited to the party because they don’t contribute and don’t do what everyone told them to do.

Children who occupied higher positions within the hierarchal social status were considerate and responsive to the group’s needs. When the group required their cooperation, they readily obliged and exhibited attentiveness. Conversely, the less popular children exhibited a lack of cooperation, disregarding the group’s needs. Those children were perceived as self-centered, concentrating on personal achievements and aspirations and failing to function as cohesive group members. One child’s description of their behavior during soccer games encapsulated this sentiment:

It’s like he (a socially rejected kid) has a big ego. Let’s say there are three players passing the ball to each other to score a goal, and he decides not to pass to them. He just goes past them and scores the goal on his own, wanting all the cheers just for himself and not caring about the other kids. It’s annoying.

In the children’s eyes, the primary distinction between social statuses resides in the child’s central focus. Accepted children centered their attention on the group, whereas unaccepted children focused on personal achievements and seeking recognition.

I think the “cool” kids really care about their friends and sports, while the “nerds” don’t really care about friends or sports. They want to be the best in school, and they want the teachers to like them the most.

### 3.2. “A ‘Cool’ Kid Wears a Brand, and Everyone’s following Him”

The second theme related to hierarchal social status during preadolescence involved children behaving according to social norms. In the eyes of the latency-age participants, behaving according to their class’s accepted social norms was desirable and important for strengthening social cohesion in the group. When the entire group behaved according to the same social norms, they focused on collective behavior.

The accepted children contributed in that way. When they behaved according to social norms, they enticed other children to behave according to those norms. As one child described, “A ‘cool’ kid wears a brand, and everyone’s following him.” This child described a situation where a popular child set a fashion trend, and the other children imitated him and behaved similarly. In other words, the children higher in the hierarchy contributed to social cohesion by promoting the children’s collective behavior according to social norms established in the classroom.

How did the popular children get the class to behave according to social norms? Jonathan (pseudonym) explained,

When you’re with the popular kids, you have power because they’re the best. Then the kids who study all the time and put a lot of effort in studies, they’re just regular kids. So they say, “Come on! Let’s hang out with the ones who live cool lives.”

Jonathan’s words demonstrate that the attraction to popular children arises because the other children admire and idolize their behavior. They perceive the popular group as powerful individuals who enjoy and add value to their lives, who “have cool lives”—and want to join them. In contrast, children who do not conform to social norms but listen to their teachers and focus on learning are labeled “ordinary,” lacking uniqueness, and are “just regular kids.”

Another child offered an explanation for the appreciation of the behavior of popular children:

They said that kids who prefer to study are “not popular,” and those who make more noise and disrupt the teachers in class are more popular. At least in our class, this happens because…. the popular kids do something special compared to everyone else. It’s a more exciting life with adventure…. They make noise, so they’re considered something better…. All the kids in the class say, “Wow, he’s a real hero. He makes noise and isn’t afraid of the teacher, so maybe it’s worth being with this hero.”

These words can shed light on how the children who participated in the study thought. The popular children in the classroom behaved in a manner reminiscent of rebellion. They acted contrary to adult moral standards (not listening to the teacher, not maintaining quiet during lessons, not studying). The classmates who observed them reacted with high emotional arousal. They had been socialized and accustomed to behaving according to adult moral standards. The popular children’s violation of these moral standards by social norms, adopting a stance against the significant adults, aroused wonder, excitement, and a sense of adventure: “Life is more exciting with adventure.” The popular children who behaved according to social norms that entailed independent group behavior—not conforming to adult morality—perceived themselves as heroic, strong, and daring. A girl in the research addressed those feelings:

They’ll not listen to the teacher, and they won’t listen to their parents, and that encourages them because others will say to them, “Wow, that’s cool! You didn’t listen to the teacher or to your parents! You’re a hero!” (Ruth)

Ruth’s words show that her admiration for the popular kids arose not only because they conformed to the social norms set by the class, but also because of the popular children’s ability to assert themselves in front of adults. This seems related to the children’s need to attach to a strong, independent group. As one of the children described,

The kids, you know, they mess around with each other and make fun, do not-so-nice things in class because it’s like they’re becoming closer friends that way. They laugh with each other, and it kinda makes them together.

That child’s words show that the children were aware they did not always behave according to adult expectations. They knew they did “not-so-nice things”. However, those actions served a purpose. They gave the children a sense that their friendships were strengthening. They felt more connected and bonded, which strengthened the group.

The hierarchal social status reinforced behavior according to social norms, weakening obedient behavior toward adults. Children who were accepted in the group conformed to the classroom social norms, even when adults did not accept those norms. On the other hand, the children who continued to obey adults lost social status, sometimes facing social exclusion. “If you wanna fit in, you can’t be the teacher’s pet. If you’re the good kid, you’ll be a ‘nerd,’ and they [the children of the class] won’t like you”.

The typical understanding of this behavior was quite binary and divided. The children could not follow the rules set by their peers and those set by adults at the same time. Thus, if their behavior did not align strictly with social norms but instead followed adult rules, their social standing diminished, often leading to exclusion.

### 3.3. “I Made the Biggest Change in My Life”

The participants’ statements reveal that the hierarchal social status was dynamic, fluid, capable of rapid change, and shifted in response to social–moral behavior. The students’ social contributions strengthened the bonds among the students in the group and their social norms. When the children were connected to a group and adhered to social norms, they ascended the social ladder within the framework. Conversely, when they were not connected to a group, they prioritized their individual preferences while disregarding the group’s needs. They preferred to heed adult moral guidance and resist the class’s social norms, leading to a decline in their social status. Children who were socially “accepted” could experience a decline in their social status. They shifted toward social exclusion when they altered their priorities, giving more weight to the adults’ guidance and values than to their peers’ companionship and norms.

Once, I was a very socially accepted child…. I was the kind who told everyone what to do, and everyone followed my lead. I didn’t treat other kids nicely. I belittled them. I was not a good student. I didn’t listen to the teachers. I talked during lessons, did things that weren’t right. I spoke rudely to other kids and picked on them. All the kids started behaving badly too, like me, …turned on each other and disrupted the teacher.

Then I promised my parents and the teacher I would change. That’s when I made the biggest change in my life. Because I changed and became a better student. I started listening to the teacher in class, talking less, and stopped doing things that weren’t right. But the other kids decided to turn against me. My class status was getting worse. I really felt it, and I would cry during breaks. In class, they annoyed me and talked behind my back. When I was alone, I just went crazy. It seemed like the end of the world to me.

This girl, once the “class queen”, became socially rejected. She described a situation where her parents and teachers exerted heavy pressure on her to change her bad behavior. After she shifted from conforming to the class’s social norms to adult rules, the children in her class ostracized her. Her choice came at a significant cost. The children no longer remembered her as the “class queen” and shunned her, causing her great suffering.

A reversal in social status within the classroom can also occur when a socially rejected child alters their behavior and adheres to social norms. This behavior change can elevate their social status. “Perhaps this boy…was ostracized, and then he somehow managed to return to being a normal boy, and so they want to accept him.” These words are from a child who explained how a socially rejected child could change their social status. Their social status would improve when they returned to “normal” behavior—accepted and conventional in the other children’s eyes.

## 4. Discussion

Inequality in hierarchical social status, especially among socially excluded children, profoundly affects preadolescents’ [17,18,19], physical health [23,24,25,27,48,49], emotional health [28,29,30], and cognition [24,31,66]. The historical view of the hierarchy—through a lens of moral deficiencies—challenges the education system’s approach and interventions [67]. Instead, developing a frame of reference that allows adults, educators, therapists, and researchers to handle this serious phenomenon more effectively is essential. Thus, this study investigated the phenomenon of hierarchal social status in preadolescence from a developmental perspective. Three themes related to the research objectives were identified from the participants’ statements:Accepted children are better friends;A “cool” kid wears a brand, and everyone’s following him;I made the biggest change in my life

These themes provide valuable insights into the dual contribution of hierarchal social status—at the group and individual levels. At the group level, it strengthens social bonds and enhances the classroom’s social cohesion. Children need to belong to, reinforce, and integrate within the group. Their accounts make evident how hierarchal social status relates to social norms, fostering group unity and cohesiveness.

At the individual level, hierarchal social status plays a role in developing the social skills necessary for group membership. When children look up to those higher in the hierarchal social status and aspire to emulate them, they join collective efforts while making individual changes. On the one hand, they cultivate their social senses, aiding their integration into the group as meaningful and helpful members who contribute to the group’s well-being. On the other hand, they adopt behaviors that consider the needs of others, reducing exclusive and egocentric attitudes.

This research offers a fresh perspective on the normative developmental aspect of hierarchical social status. Previous studies often suggested that children’s hierarchies are driven primarily by the pursuit of power and control [43,45,46], believing that bullying and social rejection are strategic means to those ends [51].

Contrary to these notions, our findings indicate that children are not motivated primarily by personal desires for popularity or affection [9,10]. Instead, they engage in a developmental challenge to form a united, cohesive group and acquire skills to be cooperative and valuable group members. Socially accepted children do not fixate on power struggles or the quest for popularity and love. They gain social acceptance by demonstrating care for the group and promoting its cohesion and unity. They catalyze the reintegration of children who deviate from the group, act primarily in their self-interest, or defy social norms within the classroom. It can be argued that children in positions of a higher hierarchical social status actively contribute to the group’s welfare and show attentiveness to others’ needs. This premise challenges the notion that those at the top of the hierarchy are motivated solely by self-gain.

Another innovation of this study is that it challenges the notions of rigid, unchanging social status positions in the classroom, or fixed personality profiles for children in different positions (e.g., [36,37,38]). Furthermore, it highlights the dynamic nature of hierarchal social status in preadolescence, showing that social status can change based on efforts made for the group’s benefit. This study’s findings underscore the potential for social status to undergo significant changes contingent upon children’s behavior. When children prioritize group camaraderie, act in the group’s best interests, and adhere to social norms, they ascend in hierarchal social status. Conversely, a child who follows parental morals while holding a high position in the hierarchal social status may experience a decline in their social standing.

This research provides a more comprehensive understanding, shedding light on diverse findings from studies exploring children’s strategies for acquiring power, including aggression [51,53,54] and prosocial behaviors [52,55,56,57,58,68,69]. Our research findings reveal that aggressive and prosocial behaviors are not isolated strategies to gain power and control within the group. Instead, they reflect the prevailing societal norms within specific classroom settings.

Children have the agency to choose to adhere to aggressive or prosocial norms. In classrooms where aggressive norms are dominant, children are inclined to conform to these norms. In contrast, in classrooms where prosocial norms prevail, children are motivated to adhere to those norms. Notably, although these norms differ between classrooms, social status is constantly determined by how much children align their behavior with the prevailing social norms, whether it be aggressive or prosocial.

The findings of this research align closely with the latency socio-emotional learning developmental theory [70,71,72,73,74,75,76]. This theory suggests that the latency phase involves the evolution of children’s social selves [77,78,79,80]. Traditionally, the social self has been defined as the narrow part related to social group membership, connections [77], or a person’s characteristic behavior in social situations that may differ from their real self [81]. Based on this study’s findings about latency-stage children, this term could be redefined to emphasize that the elementary school child’s self is fundamentally social—and their self-development is intimately linked to the social context. (This represents merely the developmental phase of the latency age; as adolescents emerge, a notable shift occurs. The construction of the self encompasses the addition of the “social self”, alongside various other components of the self).

This implies that developing a social self during latency necessitates a real, external societal environment—specifically, an active classroom community where social norms function as a cohesive entity. The social self’s development requires not only social norms, uniformity, and defiance of adults, but also the presence of hierarchical social status. A recent quantitative study in fourth- and fifth-grade classrooms supported this new approach. That study found that high social status (i.e., being well liked and accepted) correlated with high social norms and reduced egocentrism. The more the children were connected to the group, had high social status, and could move flexibly with the group’s social norms, the lower their scores for egocentrism and loneliness.

The findings of this research align with previous studies. Children’s adherence to social norms is crucial at this age and takes precedence over their adherence to adult morality [82]. Adherence to social norms contributes to the formation of a cohesive group identity [83] and collective entity [84].

Children of low social status were in a more egocentric position, seeing themselves at the center and top of the social ladder while noticing a hierarchal social status that eliminated their centrality [75]. This study’s findings suggest that investing time and resources in interventions to combat social hierarchy inequality and bridge social status gaps might be less effective than preventive educational interventions to facilitate this age group’s developmental challenges.

Per those research findings, an intervention program called “From Me to We” was recently developed in Israel for grades 4 and 5 to reduce that social status gap. Children enrolled in the program, initially in the “Me” position, characterized by a rigid, self-centered perspective often leading to declines in social status, undergo a transformative process to a “We” position. Their egocentric attitudes diminish, and they connect with the group while embracing its social norms, leading to increased social status. The intervention has demonstrated remarkable effectiveness in enhancing class cohesion, reducing violence and bullying, and elevating the hierarchical social status of all children, especially those who had been socially marginalized or rejected.

### 4.1. Study Limitations and Directions for Future Research

This study’s qualitative methodology was intended to offer a unique preliminary insight into children’s perceptions. However, it also presents several limitations that warrant consideration. First, the sample was recruited using the snowball method, suggesting that the findings should be validated later using randomly selected large-scale representative samples. Second, due to ethical concerns, this preliminary exploration did not distinguish subgroups of children who may demonstrate specific roles within peer-rejection interactions. Therefore, future research should investigate whether outcomes were linked to the respondents’ roles as rejectors, rejected individuals, or bystanders (as self- or teacher-reported). Third, the children were asked about their *parents’* responses instead of their *teachers’* responses, due to the potential influence of the teachers’ presence on their expression. This aspect should be reconsidered in future studies to account for the possible impact of teachers’ presence. Last, further research should delve into cultural, religious, gender, and generalization issues.

Expanding on these considerations, it is crucial to address additional methodological intricacies regarding focus group dynamics, which also serve as limitations. The focus group is a tool that requires three essential elements: homogeneity within groups, avoidance of situations that foster incommunicability, and an existence limited to the present moment without a past or future. *Homogeneity within groups* implies that focus groups should not mix the most popular and most rejected children, as this could lead to the domination of the discussion by the most popular children. Consequently, this paper may predominantly reflect the viewpoints of the most popular children. *Avoiding situations that foster incommunicability* is similarly critical. These situations occur when the presence of an authority figure, such as a teacher or group leader, influences what others express. The presence of a teacher may have influenced the discourse of the most popular children, whereas a hierarchical structure within the group could hinder open expression from those lower in the hierarchy. Further, focus groups should only exist in the present moment, devoid of past or future interactions among participants. When participants have prior acquaintances or future encounters, their ability to express themselves freely may be compromised.

These elements should be considered in future research that delves deeper into the subject than this preliminary study. Other areas to consider for further research include the interconnections between hierarchal social status and responsiveness and/or aggressiveness.

### 4.2. Conclusions

This study delved into the world of social status hierarchy among latency-age children. It enhanced our understanding and shed light on the normative developmental aspect of social status hierarchy during this crucial phase. Beyond understanding the practical implications, this study illuminates a promising outlook. It underscores that children traverse a distinctive juncture in their social development when they reach the point at which social status hierarchy holds significant sway. Children at the top of the social hierarchy are not motivated by personal desires for popularity, power, control, affection, or self-gain. Instead, they care for the group, contribute to the group’s welfare, and promote cohesion and group unity. Further, the social hierarchy in preadolescence is not fixed; rather, it is dynamic and subject to change. These newfound insights spark a unique, re-envisioned perspective on social status dynamics, ushering in opportunities for positive metamorphosis and improvements in the lives of children as they grapple with the intricacies of social hierarchy.

## Figures and Tables

**Table 1 children-11-00547-t001:** Focus group composition.

Focus Group	Number of				
	Participants	“Most Popular” ^a^	“Most Rejected” ^a^	Boys	Girls
1	9	3	2	0	9
2	20	7	3	9	11
3	7	5	1	0	7
4	16	2	2	4	12
5	17	5	4	10	7
6	7	2	1	0	7
7	10	7	1	6	4
8	23	8	5	19	4
9	19	4	2	10	9
10	5	0	1	3	2
11	4	0	1	1	3
12	3	3	0	3	0

Note. ^a^ Most popular” and “most rejected” children in the class according to the attending homeroom teacher.

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
