# Peer review of "“A ‘Cool’ Kid Wears a Brand, and Everyone’s following Him” Hierarchal Social Status in Preadolescence: A New Developmental Perspective"

_children, 2024, doi:10.3390/children11050547_

Round 1

Reviewer 1 Report

Comments and Suggestions for Authors

The topic of studying the effects of inequalities in social status on adolescence seems to me to be a timely topic.

The summary must include the context, the region, the country where the study was carried out

I think the Approach method section, the participants, the procedure, and the ethics section are well described.

However, I think that the analysis section is still good, since the authors point out that they have followed Van Kaam's method and perfected by Moustakas (1994) and they describe the steps very well "To improve the reliability of the findings, .... .

A professional colleague of the researcher volunteered to be the second reader, and both referred.

Seen all the dialogues in the focus groups. The process included seven stages,....." the authors must explain that this is done in order to give credibility to the qualitative study, which requires information triangulation processes (informants, researchers,.... )

It would be interesting to also see, in addition to the major themes or categories in the results

"The thematic analysis of participants' responses to the research story yielded three overarching themes: (a) Accepted children are better friends,” (b) A “cool” kid wears brand, and everyone's following him, and (c) I made the biggest change in my life. what could be the three big categories" know the subtopics, or subcategories. For this reason, a table of appearances of the results, with percentages, frequencies, and conclusions, would be of great interest.

Sánchez, M. B., Sáez, P., Gil-Madrona, M., y Martínez M. (2021). Desarrollo psicomotor y su vinculación con la motivación hacia el aprendizaje y el rendimiento académico en Educación Infantil. Revista de Educación, 392(55).

Contreras Jordán, O. R., González-Martí, I., & Gil-Madrona, P. (2019). The difficulty of implementing teaching-based competencies in Spain. Education Policy Analysis Archives27, 121. https://doi.org/10.14507/epaa.27.4053

I ask the authors to review and incorporate into their work the procedure used in these works.

Gil-Madrona, P. Prieto-Ayuso, A., Dos Santos Silva, SA, Serra-Olivares, J., Aguilar Jurado, MA & Díaz-Suárez, A. (2019). Hábitos y comportamientos relacionados con la salud de los adolescentes en su tiempo de ocio. Anales de psicología, 35(1),140-147. http://dx.doi.org/10.6018/analesps.35.1.301611 DOI: https://doi.org/10.6018/analesps.35.1.301611

Gil-Madrona, P., Aguilar-Jurado, M. Á., Honrubia-Montesinos, C. & López-Sánchez, G. (2019). Physical Activity and Health Habits of 17- to 25-Year-Old Young People during Their Free Time. Sustainability, 11(23), 1-13. https://doi.org/10.3390/su11236577

In the topic of study, this other work can provide information about adolescence.

Author Response

Please see the response in the attached file. Thank you.

Reviewer 2 Report

Comments and Suggestions for Authors

I would like to commend the authors for their insightful exploration of  a comprehensive psychological developmental understanding of hierarchal social status during the latency phase. My suggestions for revision are minor refinements aimed at enhancing clarity.

Introduction:

- Consider specifying social factors that play a significant role in preadolescence.

- What are the associations between social skills and aggressiveness.

− Hypotheses should be formulated in addition to research questions.

Methods:

- Consider providing a more comprehensive overview of the sample.

- Provide additional context or rationale for the distribution of children within the sample. For instance, discuss whether this distribution aligns with the broader demographic profile of the country.

Findings, discussion and conclusions:

- The finding regarding social norms could be broader described as it is a very interesting result (p. 7).

- When discussing the associations between the hierarchal social status and responsiveness (p. 10-11), consider elaborating on how they are interconnected.

- Can you extend the following way of thinking: “This implies that developing a social self during latency necessitates a real, external societal environment”? (p. 10).

Author Response

(The authors gave the same response as above.)

Reviewer 3 Report

Comments and Suggestions for Authors

The paper entitled: “A ‘Cool’ Kid Wears a Brand, and Everyone’s Following Him” Hierarchal Social Status in Preadolescence: A New Developmental Perspective” examines the phenomenon of hierarchal social status in preadolescence to address social status disparities. This study concludes that children at the top of the hierarchy are not motivated by personal desires for popularity or affection. They are not motivated by self-gain. On the contrary, they care for the group, contribute to the group’s welfare, and promote cohesion and unity. These findings contradict the broadly accepted idea that children at the top of the hierarchy are driven primarily by the pursuit of power and control. On the other hand, the idea of the dynamic nature of hierarchal social status in preadolescence also challenges the broadly accepted idea that social status positions in preadolescence are unchanging.

According to these results an intervention program called “From Me to We” was developed in Israel to reduce violence and bullying in the classroom and enhance class cohesion. The “Me position” is an egocentric position portrayed by children of low social status. Also, children lower in the hierarchy obey adults, listen to the teacher and to their parents, etc. On the contrary, the “We position” is a reduced egocentrism position portrayed by children higher in the hierarchy. Also, children of high social status act contrary to adult moral standards. Id est: make noise in the class, disrupt the teachers, etc.

At the methodological level, this paper is based on a qualitative approach. 140 Jewish Israeli 5th-grade children aged 10 to 11 years from 12 classes from five elementary schools participated in 12 focus groups. In each focus group participated children from the same class. I agree with the author: in-depth interviews were inappropriate because of the children’s age. Therefore, the focus group is a more appropriate research tool to study children aged 10 to 11 years. Also, the way the author guided the discussion was correct. As the author said: “dialogue was guided toward general attitudes or perceptions without delving into specific classroom occurrences. The core of each focus group session revolved around discussing “other” children within a specially crafted story about children in another city” (p. 4).

However, at the methodological level, this study contains important weaknesses that were not properly addressed in the Study Limitations and Directions for Future Research section. For example, the author recognizes that “due to ethical concerns, this preliminary exploration did not separate subgroups of children”. However, this is far from being the only limitation of this study.

The focus group is a tool that requires 3 things: 1. Groups to be homogeneous, 2. Avoid situations that generate incommunicability, 3. Groups should only exist in the present, they should not have a past or a future.

1.      Groups to be homogeneous, means that focus groups should not mix the most popular and most rejected children in the same focus group because most popular children will monopolize and control the discussion. Therefore, I think that this paper reflects principally the view of the most popular children.

2.      Avoid situations that generate incommunicability. A situation that generates incommunicability is when the presence of somebody (for example a teacher, or a person who is the leader of the group) influences what another person says. In my opinion, the presence of the teacher had some influence on the most popular children's discourse. Also, the presence of a leader of the group could influence the discourse of the children lower in the hierarchy, because people in the hierarchy are afraid to disagree with people higher in the hierarchy.

3.      Groups should only exist in the present, they should not have a past or a future, which means that the children participating in a focus group will meet the other children participating in a focus group only in the present, not in the past (id est: they didn’t meet them before) nor in the future (id est: they won’t meet them in the future). When people participating in a focus group are acquaintances or colleagues, they are not free to speak openly.

I know that conducting focus groups with children makes it very difficult to follow these rules because of ethical concerns. However, the author must explain this in detail in the Study Limitations and Directions for Future Research section.

This is my concern about this paper.

Author Response

(The authors gave the same response as above.)
